# A New Tailored Nanodroplet Carrier of Astaxanthin Can Improve Its Pharmacokinetic Profile and Antioxidant and Anti-Inflammatory Efficacies

**DOI:** 10.3390/ijms25147861

**Published:** 2024-07-18

**Authors:** Kumudesh Mishra, Nadin Khatib, Dinorah Barasch, Pradeep Kumar, Sharon Garti, Nissim Garti, Or Kakhlon

**Affiliations:** 1Department of Neurology, The Agnes Ginges Center for Human Neurogenetics, Hadassah-Hebrew University Medical Center, Jerusalem 9112001, Israel; kumudeshmishra@gmail.com (K.M.); kumar.pradeep@mail.huji.ac.il (P.K.); 2Lyotropic Delivery Systems Ltd., Hi-Tech Park, Row 5(1), Edmond J. Safra Campus, Jerusalem 9139002, Israel; nadin@lds-biotech.com (N.K.); sharon@lds-biotech.com (S.G.); 3Mass Spectrometry Unit, Institute for Drug Research, School of Pharmacy, Hebrew University of Jerusalem, Jerusalem 9112102, Israel; 4Faculty of Medicine, Hebrew University of Jerusalem, Jerusalem 9112001, Israel

**Keywords:** astaxanthin, nanodroplet formulations, oxidative damage

## Abstract

Astaxanthin (ATX) is a carotenoid nutraceutical with poor bioavailability due to its high lipophilicity. We tested a new tailored nanodroplet capable of solubilizing ATX in an oil-in-water micro-environment (LDS-ATX) for its capacity to improve the ATX pharmacokinetic profile and therapeutic efficacy. We used liquid chromatography tandem mass spectrometry (LC-MS/MS) to profile the pharmacokinetics of ATX and LDS-ATX, superoxide mutase (SOD) activity to determine their antioxidant capacity, protein carbonylation and lipid peroxidation to compare their basal and lipopolysaccharide (LPS)-induced oxidative damage, and ELISA-based detection of IL-2 and IFN-γ to determine their anti-inflammatory capacity. ATX and LDS-ATX corrected only LPS-induced SOD inhibition and oxidative damage. SOD activity was restored only by LDS-ATX in the liver and brain and by both ATX and LDS-ATX in muscle. While in the liver and muscle, LDS-ATX attenuated oxidative damage to proteins and lipids better than ATX; only oxidative damage to lipids was preferably corrected by LDS-ATX in the brain. IL-2 and IFN-γ pro-inflammatory response was corrected by LDS-ATX and not ATX in the liver and brain, but in muscle, the IL-2 response was not corrected and the IFN-γ response was mitigated by both. These results strongly suggest an organ-dependent improvement of ATX bioavailability and efficacy by the LDS-ATX nanoformulation.

## 1. Introduction

Astaxanthin (ATX, β-carotene-4,4′-dione, trans-Astaxanthin), a xanthophyll carotenoid, is a red dietary carotenoid nutraceutical isolated from *Haematococcus pluvialis*. ATX is an antioxidant and anti-inflammatory FDA-approved supplement. The antioxidant capacity of ATX is ascribed to both its chemical activity as a quencher of singlet oxygen (^1^O_2_) and free radical scavenger [1], and as an activator of nuclear factor erythroid 2-related factor 2 (Nrf2) and its downstream antioxidant genes [2,3,4,5,6,7]. This antioxidant capacity has been demonstrated in numerous pre-clinical and clinical studies such as protection against aging-associated oxidative damage [8] and ischemia-reperfusion damage [9]. ATX also has an anti-inflammatory capacity. This capacity is mainly ascribed to the suppression of the nuclear translocation of nuclear factor kappa-light-chain enhancer of activated B cells (NF-κB), which suppresses the expression of pro-inflammatory cytokines. ATX has also been demonstrated to inhibit the production of the pro-inflammatory cytokines interleukin-2 (IL-2) and interferon-γ (IFN-γ) [10]. The anti-inflammatory effects of ATX are manifested, for instance, in the suppression of neuroinflammation [11] and acute inflammatory state in the liver [12].

Various reports (e.g., [13,14]) have demonstrated ATX’s safety and efficacy in counteracting aging-related disorders, diabetes, neurodegenerative disorders, cardiovascular diseases, immunodeficiency, Chagas disease, and more. All of these beneficial effects are attributed to the relatively strong antioxidant capacity of ATX. However, the natural form of ATX is poorly bioavailable as an orally delivered substance [15], and as a result, it requires a specific delivery system. We believe that the potency of ATX can be significantly improved by increasing its bioavailability through the use of tailored nanodroplets, which have already been shown to improve the efficacy of other substances (e.g., [16,17]). The aim of this study was therefore to test whether a tailored nanodroplet formulation of ATX (LDS-ATX), designed and manufactured by Lyotropic Delivery Systems (LDS) Ltd., is capable of improving the antioxidant and anti-inflammatory capacities of ATX in vivo, as manifested by protection against lipopolysaccharide (LPS)-induced oxidative and inflammatory injuries [18,19]. To this end, we used wild type mice in which oxidative and inflammatory injuries were induced by LPS and assessed by antioxidant, protein carbonylation, and lipid peroxidation as well as anti-inflammatory assays on isolated tissues. Our results demonstrate that LDS-ATX is superior to oleoresin-dispersed ATX (AXT) in both improving bioavailability and reducing LPS-induced pro-oxidant status, inflammation, and oxidative damage to both proteins and lipids in the murine brain, liver, and muscle tissues.

## 2. Results

### 2.1. Pharmacokinetics of LDS-ATX vs. ATX

To determine the serum concentrations of ATX, plasma from untreated mice was spiked with different concentrations of ATX. The standard curve thus generated (Figure 1) was used to interpolate ATX concentrations in the plasma. Our results (Figure 2) demonstrate that solubilizing ATX into the tailored nanodroplet formulation LDS-ATX led to a significant accumulation of the drug in plasma over time. In addition, the half-life (t_max_) of ATX in the plasma was delayed from 30 min in oleoresin ATX to 60 min in LDS-ATX. These two results strongly suggest that the bioavailability of ATX is significantly increased by the tailored nanodroplet formulation LDS-ATX compared to the classical commercially available product dispersed in oleoresin (AXT).

### 2.2. Antioxidant Efficacy of LDS-ATX vs. ATX

The antioxidant capacity of ATX in comparison to LDS-ATX was tested in a pro-oxidant inducible system. To this end, mice were orally pretreated with ATX in oleoresin, or as LDS-ATX, for 24 h. Subsequently, oxidative stress was induced with 1 mg/kg LPS [20,21] for 4 h and then the extent of antioxidant capacity and pro-oxidant defense in the brain, liver, and skeletal muscle was determined using commercial SOD activity, protein carbonylation, and malondialdehyde kits to determine the antioxidant capacity, irreversible protein oxidation, and oxidative damage by lipid peroxidation, respectively. The experimental setup, which also included testing the anti-inflammatory capacity of LDS-ATX (Section 2.3), is schematically described in Figure 3. The effect of ATX and LDS-ATX on the antioxidant capacity, as a potential for reversing oxidative damage, was determined based on changes in the superoxide dismutase (SOD) activity. The actual prevention, rather than potential to prevent, of pro-oxidant damage was determined using protein carbonylation and malondialdehyde commercial kits to determine irreversible protein oxidation and lipid peroxidation oxidative damage, respectively. In all assays, wild type mice not induced with LPS were used to assess basal, non-induced, oxidative stress. We predicted that this oxidative stress would be significantly lower than the LPS-induced stress and therefore, based on its significantly improved pharmacokinetic profile, tested only the effect of LDS-ATX in non-LPS-induced (control) mice. Our SOD activity results (Figure 4) showed that LPS reduced this activity by approximately twofold in all organs tested, as also observed by others [22]. While in all organs LDS-AXT did not affect the basal, non LPS-induced SOD activity, in the liver, LDS-ATX significantly increased the SOD activity in LPS-induced mice compared to both the untreated and empty nanodroplet treated animals. In the brain, LDS-ATX significantly increased the SOD activity compared to the empty nanodroplet-treated mice, and in the skeletal muscle, this increase was significant when compared to the untreated mice. In general, it can be concluded that LDS-ATX is superior to non-formulated ATX in maintaining reactive oxygen species (ROS) homeostasis in all tissues. This conclusion, demonstrating improvement of the antioxidant capacity, prompted us to test whether LDS-ATX could also better mitigate the extent of irreversible oxidative damage. Our results show that indeed, LPS induction produced a more than twofold increase in both lipid peroxidation and protein oxidation in all organs tested. LDS-ATX did not affect this basal oxidative stress. Lipid peroxidation (Figure 5), presumably directly affected by the lipid soluble ATX, which is dissolved in cell membranes, was in general more affected by both ATX and LDS-ATX than protein oxidation (Figure 6). The most affected organ was the liver, which is also the most metabolically active organ. In the liver, both lipid peroxidation and protein oxidation were better ameliorated by LDS-ATX than by ATX. In the brain, lipid peroxidation was more attenuated by LDS-ATX than by ATX, but protein oxidation was attenuated to the same extent by both ATX formulations, and to a lesser extent than lipid peroxidation. In the muscle, both lipid peroxidation and protein oxidation were only significantly reduced by LDS-ATX and not by ATX.

### 2.3. Anti-Inflammatory Efficacy of LDS-ATX vs. ATX

We also tested the capacity of LDS-ATX in comparison to ATX as an anti-inflammatory agent, reflecting the documented anti-inflammatory capacity of ATX (e.g., [23]). This was done by determining the effect of LDS-ATX and ATX on the attenuation of two pro-inflammatory cytokines, IL-2 and IFN-γ. According to the literature, LPS upmodulated the IL-2 and IFN-γ levels [24,25]. However, there have been also reports of non-reponsiveness to LPS in murine, as opposed to human, cells [26]. In our hands, LPS-induced IL-2 levels in the liver and brain, but only after treating the mice with an unloaded nanodroplet as a negative control. LDS-ATX but not ATX significantly reduced the IL-2 levels in the liver and brain compared to the unloaded nanodroplet control. No significant differences in IL-2 among all treatments were observed in the skeletal muscle, possibly related to a decrease in IL-2 to nearly basal levels at 4 h after LPS induction [27]. IFN-γ, on the other hand, was induced by LPS in all organs and downmodulated by LDS-ATX, and not by ATX, in the liver and brain. In skeletal muscle, both LDS-ATX and ATX attenuated the LPS-induced IFN-γ, but only compared to LPS without the unloaded nanodroplet.

### 2.4. Organ Specificity of the LDS-ATX and ATX Effects

To determine whether the effect of ATX and LDS-ATX is organ-dependent, we analyzed the results by two-way ANOVA with organ and treatment as the two factors. Our results (Table 1) showed that the antioxidative effect of ATX and LDS-ATX, as reflected in the MDA levels, protein carbonylation, and SOD activity, was significantly different between the liver, brain, and muscle. These results are in agreement with different basal propensities to oxidative injury among the liver, brain, and muscle tissues. On the other hand, the anti-inflammatory effect of ATX and LDS-ATX are cytokine dependent: ATX and LDS-ATX differentially reduced the levels of IL-2 in the liver, brain, and muscle. However, the effect of ATX and LDS-ATX on the IFNγ levels was not significantly different among the different tissues, possibly due to an overwhelming pro-inflammatory effect of IFNγ, which was also mitigated to the same extent by ATX in all body tissues.

**Table 1 ijms-25-07861-t001:** The % and *p*-values of two-way ANOVA interaction as a source of variation.

Assay	% of Total Variation	*p*-Value	Significance
MDA (Figure 5)	13.43	<0.0001	Yes
Protein carbonylation (Figure 6)	7.280	0.0064	Yes
SOD activity (Figure 4)	6.728	0.0141	Yes
IL-2 (Figure 7)	19.22	0.0151	Yes
IFN-γ (Figure 8)	3.993	0.9339	No

## 3. Discussion

Our unique LDS-ATX formulation, as all other LDS nanodroplet formulations [16,17], is a complex blend of co-solvents, hydrophilic, and hydrophobic surfactants, proportioned in a ratio meticulously studied and optimized during the development process. This unique combination results in a substantially hydrophilic delivery system, which is a strategy not implemented in other ATX excipients developed until now [28]. Importantly, this system is capable of maintaining the desired structural arrangement in a waterless state, and also facilitates the in situ formation of these structures upon post-dilution in the gastrointestinal (GI) tract and stomach following oral administration. Here, we tried to solubilize ATX by a tailored LDS nanodroplet. Several studies using different types of formulations have demonstrated a formulation-dependent improvement in the stability and antioxidant capacity of ATX (e.g., [14,29,30,31]). However, while many of these studies have focused on cell-based assays for assessing the antioxidant capacity of the formulated ATX, few studies have compared in parallel the antioxidant and anti-inflammatory efficacies of formulated ATX in different organs and in basal and induced pro-oxidant conditions, while assessing cellular antioxidant capacity and oxidative damage to both lipids and proteins. In this respect, the comprehensive analysis of our study might better recapitulate physiological conditions and thus better predict the efficacy of LDS-ATX in humans.

Importantly, due to its unique solubilization system, as described above, the LDS-ATX formulation improves ATX stability and bioavailability but does not affect its innate antioxidative capacity. This capacity is enabled by both the reducing keto and hydroxyl moieties found on each end of ATX and by its membrane spanning structure, which rigidifies it, mitigating the infiltration of endogenous promoters of lipid peroxidation [32]. A future direction for improving the therapeutic capacity of ATX might include not only LDS-ATX as a solubilization and bioavailability booster, but also the design of novel nanodroplets loaded with a mixture of antioxidants and possibly ligand-targeted to pathogenic tissues such as immune cells. Indeed, coencapsulation of ATX with the antioxidants capsaicin and resveratrol synergized the antioxidative capacity [33].

The choice of protein carbonylation and lipid peroxidation as indicators of oxidative injury is not only comprehensive, but also more accurate: protein carbonylation is a non-enzymatic irreversible post-translational modification of proteins induced by oxidative stress [34]. Total protein carbonylation is considered an informative and reliable biomarker of oxidative stress since it reports irreparable oxidative damage to proteins [34,35]. In contrast, other oxidative stress markers report the production of reactive oxygen species, which could cause oxidative damage, but can also be repaired by the cells’ antioxidant machinery, as opposed to the irreparable damage reported by protein carbonylation. Protein carbonyls are generated by the oxidation of amino acid side chains into aldehydes and ketones. These oxidized proteins can unfold and form aggregates, which are resistant to proteolysis and propagate cellular oxidative damage by mediating proteasome inhibition, thus impeding protein quality control [35].

The lipid peroxidation assay detects the oxidation of mainly cell membrane-associated polyunsaturated fatty acids, which are ultimately peroxidized to malondialdehyde (MDA), which is detected by the assay. The anti-inflammatory properties of ATX and LDS-ATX observed here can be ascribed to the relatively high abundance of polyunsaturated fatty acids in the plasma membrane of immune cells [36], causing them to generate relatively more ROS and be prone to oxidative damage more than other cell types, and thus be relatively more responsive to antioxidative therapy. Thus, an immune cell-inclined antioxidative effect of LDS-ATX and ATX can enhance the immune response, upgrading the anti-inflammatory response. The ATX specific anti-inflammatory effect is also attributed to its suppression of Nrf2 and NFκB signaling [2,3,4,5,6,7]. As our data showed, in general, the most prominent anti-inflammatory boosting, or pro-inflammatory suppression effects of LDS-ATX were found in the liver, in which pro-inflammatory cytokines are probably produced by the ATX-responsive immune cells: migrating CD4 T cells [37] or cell-autonomous liver resident T cells.

Finally, differences among the different organs in oxidative injury and its repair by LDS-ATX might result from various reasons, and more in-depth physiological and ex vivo investigation might be required to elucidate these differences. However, some suggestions can be presented here. First, perfusion of the different organs is different, with the liver being the most perfused organ, and skeletal muscle probably being the least. These differences in perfusion probably explain why, in general, muscle was less influenced by both ATX and LDS-ATX compared to the liver, and to a lesser extent compared to the brain. Another difference among the different organs tested was in the relative membrane protein contents. This difference might account for the fact that in the liver, ATX and LDS-ATX reduced both protein and membrane oxidation, while in the brain, protein oxidation was influenced less than membrane oxidation. If the relative abundance of membrane associated proteins such as transporters or channels is higher in the liver than in the brain, we can conject that MDA peroxidation will affect juxtaposing proteins more in the liver than in the brain.

## 4. Materials and Methods

### 4.1. Preparation of Tailored Nanodroplets for Astaxanthin Delivery (LDS-ATX)

Our oil-in-water confidential nanoformulation for ATX was based on a set of food-grade quality ingredients permitted for oral use as food supplements. These ingredients self-assemble into nanodomains by their simple mixing. Once all the ingredients and ATX are mixed, a clear solution-like system is formed, which is the final product. This tailored nanoformulation was developed by LDS to solubilize ATX. The maximum loading capacity was 1.0 wt%. This careful selection of ingredients underscores the commitment to ensure the highest standard of safety and quality in the development process.

The preparation of LDS-ATX involves a two-step process. Initially, empty concentrates were prepared by accurately weighing all the excipients and mixing them together with the aid of magnetic stirring plates. The concentrates were mixed while heated, using a water bath maintained at no more than 60 °C. Once the solution achieved homogeneity (clear, transparent concentrate), the active compound, ATX in oleoresin (10 wt%, Algatechnologies, Ketura, Israel), was introduced under nitrogen at a specified concentration, as presented in Figure 9. Subsequently, samples were mixed under heat and maintained at 50 ± 10 °C until ATX (the active pharmaceutical ingredient, API) became fully solubilized in the concentrate. Since the formulation exhibited a dim color after adding the API, a light microscope was used to confirm complete solubility.

Nanodroplet size (Figure 10) was measured by dynamic light scattering using the Nano-S system (Malvern-Panalytical, Malvern, UK) equipped with a 633 nm laser at 173° (back scattering). The operation was conducted using the Zetasizer^®^ program (https://www.malvernpanalytical.com/en/products/product-range/zetasizer-range/zetasizer-advance-range/zetasizer-ultra?utm, accessed on 16 May 2024). The sample underwent a significant dilution process (100 mg formulation diluted in 100 mL of purified water) prior to measurement. The transparency enabled by the dilution is crucial for the effective transmission of laser light passing through the sample without being obstructed by its density or opacity during analysis.

LDS-ATX physical stability (Figure 11) was measured using a LUMiSizer^®^ dispersion analyzer (LUM, GmbH, Berlin, Germany) in 2 mm polycarbonate cuvettes at 856 nm at 25 °C. The LUMiSizer^®^ monitors light transmission through the samples while they are centrifuged horizontally. The change in transmission indicates the stability of the systems because when the transmission profiles remain constant, the samples are considered physically stable. The shelf-life of the samples can be extrapolated based on the measurement conditions using the following formula:$$\begin{array}{c} Shelf\;life=rcf\times (number\;of\;profiles)\times interval \end{array}$$

In our case, the transmission did not degrade even after 800 profiles (cycles) were run at 1000 rcf with 60 s intervals between measurements, and therefore, the extrapolated physical shelf-life of LDS-ATX is approximately 2 years.

### 4.2. Liquid Chromatography Tandem Mass Spectrometry (LC-MS/MS)

#### 4.2.1. LC-MS/MS Materials

LC/MS-grade acetonitrile (ACN), methanol, and water were purchased from Biolab Ltd. (Jerusalem, Israel). LC/MS-grade formic acid (FA) was purchased from Fisher Chemical™ Optima™ (Pittsburgh, PA, USA). Apo-8-carotenal (APO), used as internal standard (IS), was purchased from Merck (Rehovot, Israel).

#### 4.2.2. Ultra-High Performance Liquid Chromatography (UHPLC)

LC-MS/MS analyses were conducted on a Sciex (Framingham, MA, USA) Triple Quad™ 5500 mass spectrometer coupled with a Shimadzu (Kyoto, Japan) UHPLC system. The chromatographic separations were performed on a CORTECS^®^ (Waters Corp., Milford, MA, USA) column (C18, 2.7 µm particle size, 50 × 2.1 mm), protected by a VanGuard^®^ (Waters Corp., Milford, MA, USA) cartridge (CORTECS^®^ C18, 5 × 2.1 mm). The injection volume was 5 μL. The oven temperature was maintained at 40 °C, and the autosampler tray temperature was maintained at 10 °C.

Chromatographic separation was achieved using a linear gradient program at a constant flow rate of 0.4 mL/min over a total run time of 7 min. An outline of the mobile phase gradient program is summarized in Table 2.

The column effluent was diverted away from the MS during the first 0.7 min and the last min of the run. Methanol was used to wash the needle prior to each injection cycle. All samples were analyzed in duplicate.

#### 4.2.3. MS/MS Conditions

ATX and APO (IS) were detected in positive ion mode using electron spray ionization (ESI) and multiple reaction monitoring (MRM) mode of acquisition.

The molecular ions of the compounds [M + H]+ were selected in the first mass analyzer and fragmented in the collision cell followed by the detection of the products of fragmentation in the second analyzer. Their transitions are shown in Table 3.

The TurboIonspray^®^ probe temperature was set at 600 °C with the ion spray voltage at 5500 V. The curtain gas was set at 25.0 psi. The nebulizer gas (Gas 1) was set to 50 psi, the turbo heater gas (Gas 2) was set to 10 psi, and the collision gas (CAD) was set to 8 psi. The entrance potential (EP) was set at 10 V. The collision energy potentials (CE), collision cell exit potentials (CXP), and declustering potentials (DP) for the monitored transitions are given in Table 3. The dwell time was 20 ms. Data acquisition and analysis were performed using Analyst 1.6.3 software distributed by Sciex. Quantitative calibrations (0–100 ng/mL) of ATX were performed before every batch of samples using peak–area ratios (compound versus internal standard). The calibration curve (y = a + bx) was obtained by weighted (1/y) linear least-squares regression of the measured peak–area ratio (y) versus the concentration added to the plasma (x). The limit of quantification (LOQ) was 1 ng/mL for ATX. A typical chromatogram of ATX in plasma is shown in Figure 12.

### 4.3. Pharmacokinetics

To determine the distribution and kinetic parameters of ATX in blood, 4-week-old C57Bl/6 male mice were pipetted per os 10 mg/kg ATX either as a non-formulated compound dispersed in oleoresin solution, as the control, or as the tested LDS-ATX. Mice (*n* = 3/time point) were euthanized by cervical dislocation at 0, 15, 30, 60, 90, and 240 min post administration. Blood was then collected by cardiac puncture and extracted with 1:1 acetonitrile following the established guidelines. The level of ATX in plasma was analyzed by LC-MS/MS as described above.

### 4.4. In Vivo and Ex Vivo Studies

The antioxidant effect of ATX and LDS-ATX was tested using a lipopolysaccharide (LPS)-inducible system. LPS is a Gram-negative bacterial endotoxin that acts as a strong stimulator of innate immunity generating pro-inflammatory oxidative stress. For all assays, we used 8-month-old male C57Bl/6J mice (23–25 g) ordered from Harlan Laboratories (Jerusalem, Israel) and acclimated for one week in our facility prior to the experiments. Mice were fed ad lib on a standard chow diet with constant accessibility to water. Mice were kept at a 12 h light/12 h dark cycle, at 24 °C, and 40–60% humidity. We pretreated the mice per os with 10 mg/kg ATX or LDS-ATX for 24 h, and then induced oxidative stress with 1 mg/kg LPS (injected intraperitoneally) for 4 h. We subsequently extracted the brain, liver and muscle tissues from the mice: tissues (100 mg) were collected in PBS and homogenized by bead beating using 2 mm zirconium beads and the Precellys Evolution Touch homogenizer (Montigny le Bretonneux, France) set to five cycles at a speed of 4500 to 6800 rpm. In the homogenized samples, we quantified the following: SOD activity using a dedicated kit (Merck, Rehovot, Israel, Catalogue number 19160), the extent of pro-oxidant protection using a protein carbonylation ELISA kit (Abcam, Cambridge, UK, Cat. # AB-ab238536-96), lipid peroxidation (malondialdehyde) kit (Abcam, AB-ab118970-100), and IL-2 and IFN-γ using specific ELISA kits (BioLegend ELISA kits 575409 and 430101, respectively). All assays (i.e., tissue extraction and quantification) were conducted as per the manufacturer’s instructions. Mice not induced with LPS were used to assess the basal non-induced oxidative stress and whether such stress could also be corrected by ATX. The experimental layout is shown in Table 4. All mouse work was approved by the Hebrew University Faculty of Medicine IACUC (approval # 17269).

### 4.5. Statistical Analysis

All statistical analyses were performed by Prism GraphPad v. 10. *p* < 0.05 was used as the significance value for rejecting null hypotheses. One-way ANOVA with Fischer post hoc LSD tests were used for multiple comparisons (Figure 4, Figure 5, Figure 6, Figure 7 and Figure 8). Two-way ANOVA was used to test the factor interaction (Table 1).

## 5. Conclusions

In conclusion, our tailored ATX nanoformulation LDS-ATX could significantly improve the biodistribution of ATX. Lipid peroxidation was further attenuated by LDS-ATX compared to ATX in the liver, muscle, and brain, while protein oxidation was more attenuated by LDS-ATX compared to ATX only in the liver and muscle.

## Figures and Tables

**Figure 1 ijms-25-07861-f001:**
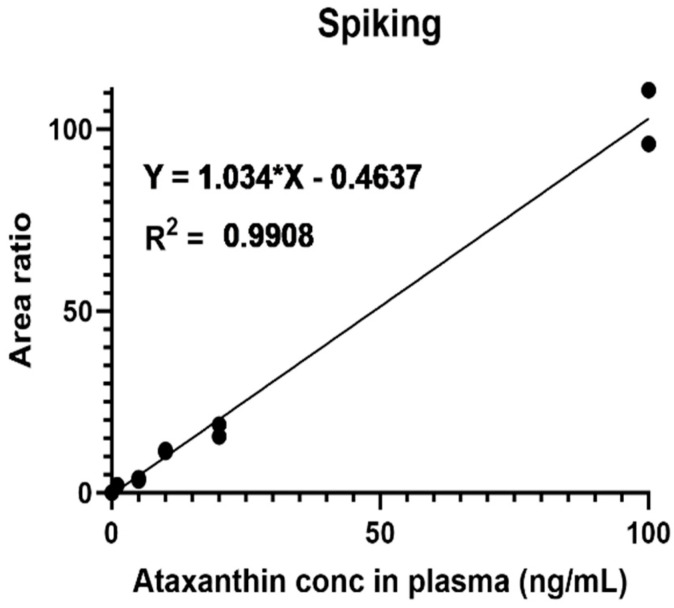
Calibration curve of the ATX levels in plasma was generated by spiking the naive plasma of *n* = 2 untreated C57Bl/6 mice with the indicated concentrations of free ATX powder.

**Figure 2 ijms-25-07861-f002:**
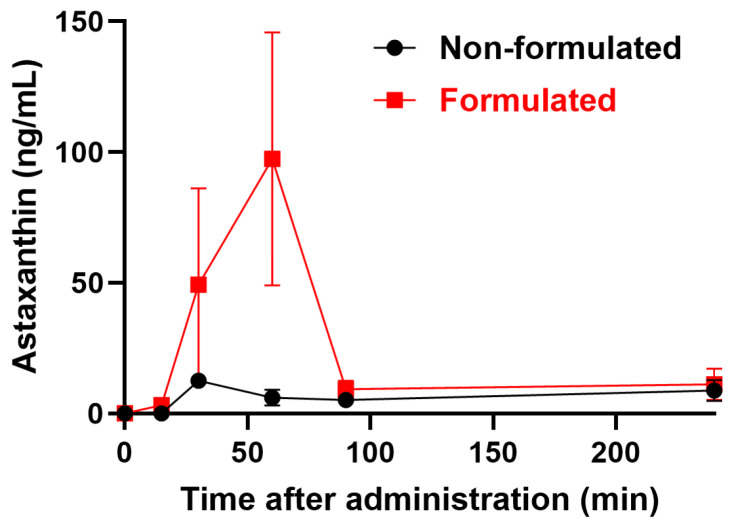
A pharmacokinetic profile (means ± SEM) of ATX in the plasma of *n* = 3/(time point) 4-week-old C57Bl/6 male mice treated with formulated (LDS-AXT) and non-formulated (AXT) astaxanthin as indicated. Area under the curve (AUC) values were 1597 ± 622 (95% confidence interval (CI): 377.9 to 2816) and 7012 ± 1882 (CI: 3323 to 10,700) for ATX and LDS-ATX, respectively. Serum levels over time of LDS-AXT were significantly higher than those of AXT, as determined by ANOVA with repeated measures (*p* < 0.05).

**Figure 3 ijms-25-07861-f003:**
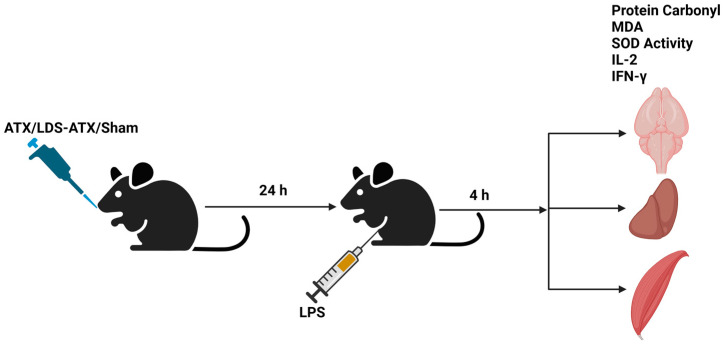
Scheme of in vivo work. Generated by BioRender.

**Figure 4 ijms-25-07861-f004:**
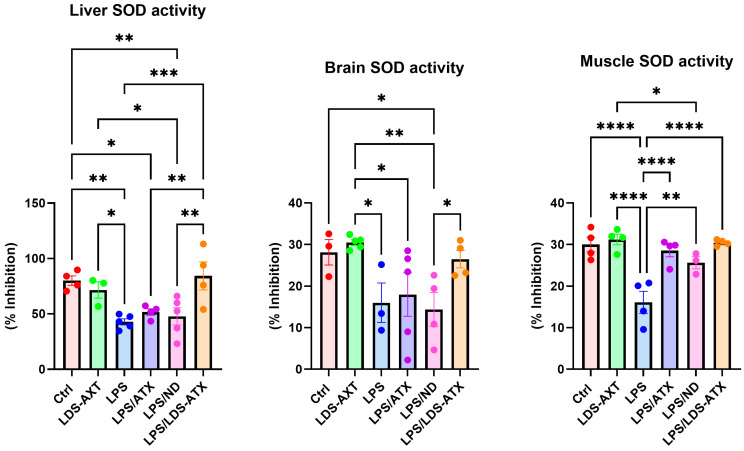
SOD activity, measured as % inhibition of ^1^O_2_ dependent WST-1 formazan formation in different tissues induced or not with LPS. SOD activity was determined by Merck’s SOD determination kit (19,160) based on the extent of the decrease in WST-1 formazan absorbance at 440 nm. Assays were performed on the liver, brain, and gastrocnemius muscle homogenates collected from *n* = 5 (outliers removed) 8-month-old C57Bl mice. The results (outliers removed) showed that LPS inhibited SOD activity, which was increased by LDS-ATX in the liver and brain and by both LDS-ATX and ATX in the muscle. Basal, non-induced SOD activity was not significantly affected by LDS-ATX. * *p* < 0.05; ** *p* < 0.01; *** *p* < 0.001; **** *p* < 0.0001. One-way ANOVA with Fisher’s post hoc test.

**Figure 5 ijms-25-07861-f005:**
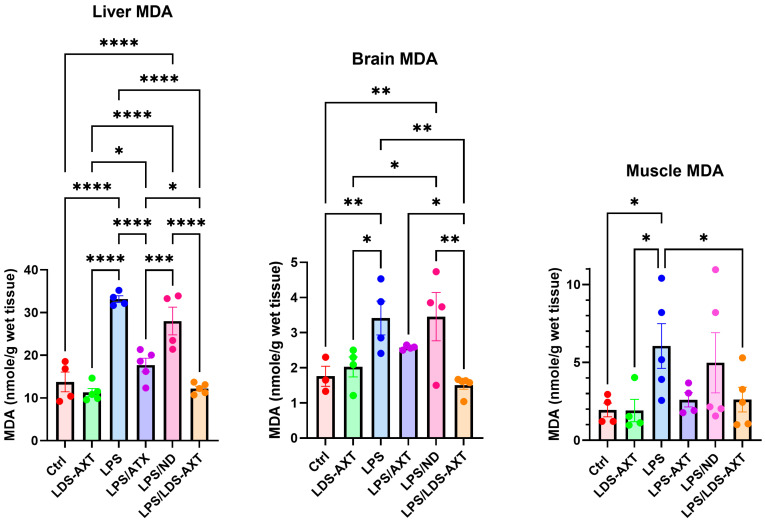
Malondialdehyde (MDA) levels in different tissues induced or not with LPS. MDA was determined by AbCam’s ab233471 commercial kit as a measure of lipid peroxidation in the liver, brain, and gastrocnemius muscle homogenates collected from *n* = 5 (outliers removed) 8-month-old C57Bl mice. The results (outliers removed) show that LPS-induced lipid peroxidation, which was attenuated by ATX and even more so by LDS-ATX. Basal, non-induced lipid peroxidation was not significantly affected by LDS-ATX. * *p* < 0.05; ** *p* < 0.01; *** *p* < 0.001; **** *p* < 0.0001; One-way ANOVA with Fisher’s post hoc test.

**Figure 6 ijms-25-07861-f006:**
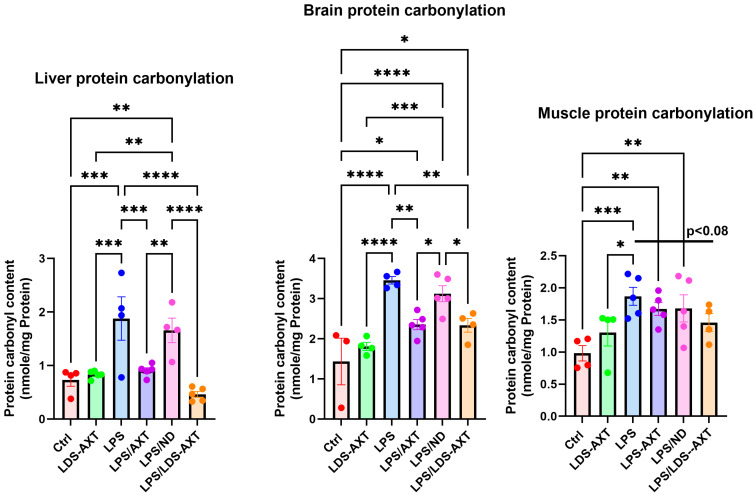
Protein carbonylation levels in different tissues induced or not with LPS. Protein carbonylation by 2,4-dinitrophenylhydrazine was determined by AbCam’s ab126287 commercial kit as a measure of protein irreversible oxidative damage in the liver, brain, and gastrocnemius muscle homogenates collected from *n* = 5 (outliers removed) 8-month-old C57Bl mice. The results (outliers removed) show that LPS-induced protein oxidation, which was attenuated by ATX, was significantly further attenuated by LDS-AXT in the liver, attenuated to the same extent by ATX and LDS-ATX in the brain, and attenuated only by LDS-AXT (*p* < 0.08) and not by AXT in the muscle. Basal, non-induced lipid peroxidation was not significantly affected by LDS-ATX. LDS-ATX inhibited protein carbonylation in LPS-induced animals only as a trend (*p* < 0.08). * *p* < 0.05; ** *p* < 0.01; *** *p* < 0.001; **** *p* < 0.0001. One-way ANOVA with Fisher’s post hoc test.

**Figure 7 ijms-25-07861-f007:**
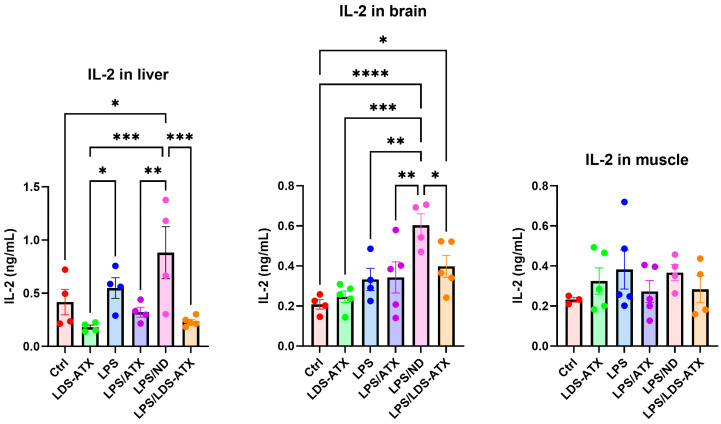
IL-2 levels in different tissues induced or not with LPS. IL-2 levels were determined by BioLegend’s (San Diego, CA, USA) ELISA 575409 kit in homogenates collected from *n* = 5 (outliers removed) 8-month-old C57Bl mice. The results showed that in LPS-induced animals (treated with an unloaded nanodroplet (LPS/ND), IL-2 was significantly increased in the liver and brain and downmodulated by LDS-ATX. No significant differences in IL-2 levels were observed among the different treatments in skeletal muscle tissue. Basal, non-induced lipid peroxidation was not significantly affected by LDS-ATX in all tissues. * *p* < 0.05; ** *p* < 0.01; *** *p* < 0.001; **** *p* < 0.0001. One-way ANOVA with Fisher’s post hoc test.

**Figure 8 ijms-25-07861-f008:**
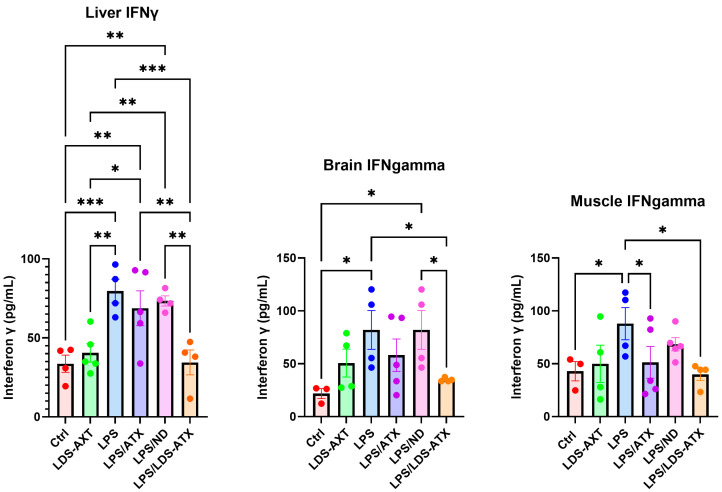
IFN-γ levels in different tissues induced or not with LPS. IFN-γ levels were determined by BioLegend’s ELISA 430101 kit in homogenates collected from *n* = 5 (outliers removed) 8-month-old C57Bl mice. The results showed that LPS significantly increased the IFN-γ levels in all tissues. LDS-ATX, but not ATX, significantly downmodulated IFN-γ in the liver and brain. In skeletal muscle, both ATX and LDS-ATX restored the IFN-γ levels, but only compared to LPS applied without an unloaded nanodroplet formulation. Basal, non-induced lipid peroxidation was not significantly affected by LDS-ATX in all tissues. * *p* < 0.05; ** *p* < 0.01; *** *p* < 0.001; One-way ANOVA with Fisher’s post hoc test.

**Figure 9 ijms-25-07861-f009:**
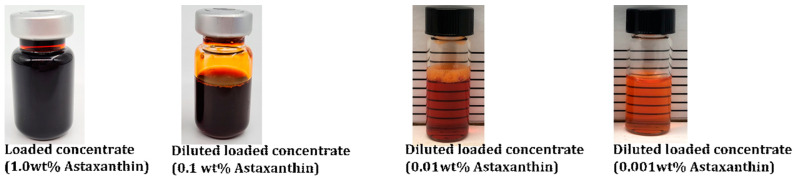
Visual appearance of the formulated loaded concentrate and its dilutions in water.

**Figure 10 ijms-25-07861-f010:**
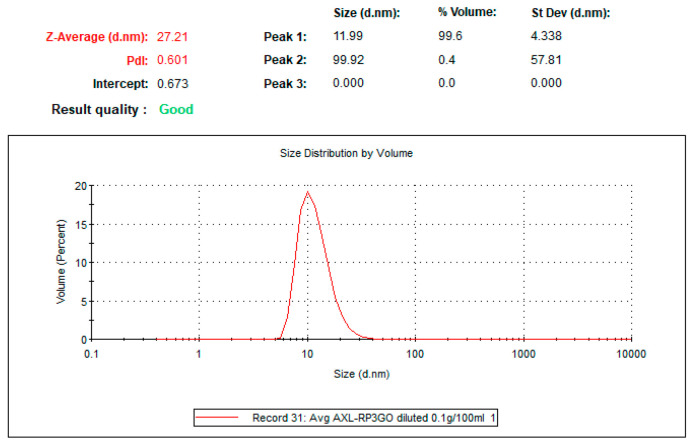
LDS-ATX droplet size determined by dynamic light scattering (DLS). Average droplet size was 11.99 nm with only very minor populations of larger-sized droplets, as can be seen in the peak distribution table on the top right. Values of other parameters (Z-Average, polydispersity index (PDI) and intercept) are indicated on the top left.

**Figure 11 ijms-25-07861-f011:**
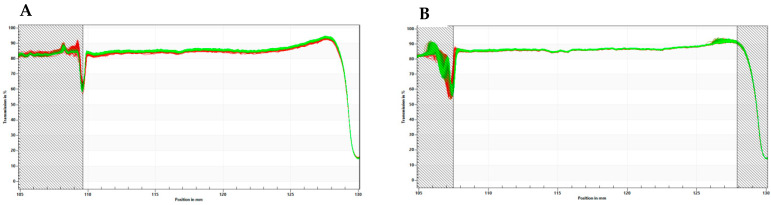
Physical stability measured by a LUMiSizer® for (**A**) concentrated system (1.0 wt% Astaxanthin) and (**B**) diluted system (0.1% Astaxanthin). Transmission profiles (*y* axis) were plotted as a function of position (*x* axis) and time (line color from red to green). No significant changes in transmission were observed in both the concentrated and diluted systems, meaning that the systems are physically stable.

**Figure 12 ijms-25-07861-f012:**
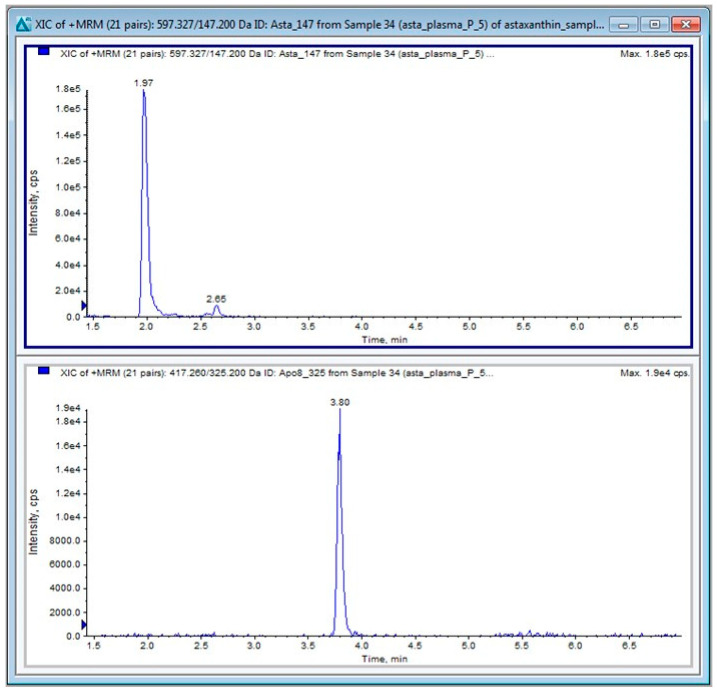
A typical chromatogram of ATX (**upper panel**), and the APO IS (**lower panel**) in a mouse plasma sample.

**Table 2 ijms-25-07861-t002:** Gradient program: solvent A is 0.1% formic acid (FA) in water and solvent B is 0.1% FA in ACN.

Time (min)	Solvent A (%)	Solvent B (%)
0.0	23	77
0.5	23	77
2.5	5	95
4.5	5	95
5.0	23	77
7.0	23	77

**Table 3 ijms-25-07861-t003:** Multiple reaction monitoring (MRM) transitions and parameters for ATX and APO (IS) in positive ion mode. *m*/*z*: mass to charge ratio; DP, declustering potential; CE, collision energy; CXP, collision cell exit potential; V, volts; eV, electron volts; Rt, retention time.

Name	Precursor (*m*/*z*)	Product (*m*/*z*)	DP (V)	CE (eV)	CXP (V)	Rt (min)
ATX	597.3	Quantifier 147.2	6	29	16	2.0
		Qualifier 173.1	11	23	10	
APO	417.2	Quantifier 325.2	16	13	12	3.8
		Qualifier 159.2	21	27	16	

**Table 4 ijms-25-07861-t004:** Experimental layout for the animal study.

LDS-ATX	ATX	Unloaded Nanodroplet	LPS	No. of Animals
+	−	−	+	5
−	−	+	+	5
−	+	−	+	5
−	−	−	+	5
+	−	−	−	5
−	−	−	−	5

## Data Availability

Data is contained within the article. The original contributions presented in the study are included in the article. Further inquiries can be directed to the corresponding authors.

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
