# Peer review of "A New Tailored Nanodroplet Carrier of Astaxanthin Can Improve Its Pharmacokinetic Profile and Antioxidant and Anti-Inflammatory Efficacies"

_ijms, 2024, doi:10.3390/ijms25147861_

Round 1

Reviewer 1 Report (Previous Reviewer 2)

Comments and Suggestions for Authors

The article of Kumudesh Mishra et al. “A New Tailored Nano-Droplet Carrier of Astaxanthin Can Improve Its Pharmacokinetic Profile and Antioxidant Efficacy” has been improved and now warrants publication in ijms.

However, there are some comments on submitting the article.

1. Figures are poor quality.

2. References 3, 13, 15, and 16 do not have DOI

3. Reference 16 is presented in different fonts.

Comments on the Quality of English Language

Minor editing of English language required

Author Response

Comment: The article of Kumudesh Mishra et al. “A New Tailored Nano-Droplet Carrier of Astaxanthin Can Improve Its Pharmacokinetic Profile and Antioxidant Efficacy” has been improved and now warrants publication in ijms.

However, there are some comments on submitting the article.

  1. Figures are poor quality.
  2. References 3, 13, 15, and 16 do not have DOI
  3. Reference 16 is presented in different fonts.

Reply: 

We thank the reviewer for finding our manuscript acceptable for publication in IJMS.

Specifically, regarding the new comments:

  1. We tried, but could not further improve the quality of our figures. We will consults with IJMS production about that prior to publication.
  2. Indeed we could not find the DOIs of the references mentioned by the reviewer. We will consult with production about that issue as well.
  3. Corrected.

Reviewer 2 Report (Previous Reviewer 3)

Comments and Suggestions for Authors

I can accept the revised manuscript.

Author Response

Comment: I can accept the revised manuscript.

Reply: We thank the reviewer for the recommendation to accept our publication. We didn't find any further comments to address.

Reviewer 3 Report (New Reviewer)

Comments and Suggestions for Authors

Hi, Kumudesh Mishra,

Thank you very much for submitting your manuscript to our journal, your paper is very good!  But I still have a couples of questions about your contents:

1. How do you synthesize the nano droplet of ATX?  What is the size and the zeta potential of nano droplet? I think you should provide the detail information in the method and result

2. For Figure 2, the error bar of formulated LDS-AXT is too big, you can increase the number of mice to 4 or 5. By the way, whether these two groups have a significant difference?

3. I really suggest you detect the biodistribution of LDS-ATX and detect the toxicity of LDS-ATX by HE staining.

4. The lager part of nano droplet of LDS-ATX usually accumulates into liver of mice, why the SOD activity of LDS-ATX is less than the control group (Figure4)?

Author Response

Comment: How do you synthesize the nano droplet of ATX?  What is the size and the zeta potential of nano droplet? I think you should provide the detail information in the method and result

Reply: Since January 2024, LDS-AXT is a commercial product sold as nutraceutical around the world. Subsequently, exact detailed information on the preparation of LDS-AXT is protected by patent law. Nevertheless, we added the following information in Methods: All ingredients (which, again, cannot be disclosed) are simply mixed together at room temperature and self-assemble into nanodomain droplets. The droplets are all formed with nonionic surfactants and therefore there is no need to measure zeta potential.  Once all the ingredients are mixed with AXT, a clear solution-like system is formed, which is the final product.

Comment: For Figure 2, the error bar of formulated LDS-AXT is too big, you can increase the number of mice to 4 or 5. By the way, whether these two groups have a significant difference?

Reply: Accumulation over time (pharmacokinetic profile) of LDS-AXT is significantly higher than that of AXT (p<0.05). This was determined by ANOVA with repeated measures, which already took into account n and the size of the error bars. We have now added this statistical analysis. Thank you for this comment. In addition, for animal ethics reasons, pharmacokinetic profiles are normally determined in n=3 animals per time point, unless the statistical significance is insufficient, which, as stated above, is not the case here. This is also the practice in other publications (e.g., Ong et al. (2016) Pharm Res 33: 563, our own publication Kakhlon et al (2021) EMBO Mol Med 13:e14554 and more).

Comment: I really suggest you detect the biodistribution of LDS-ATX and detect the toxicity of LDS-ATX by HE staining. 

Reply: As stated above, LDS-AXT is a commercial product already in the market. A comprehensive battery of all required safety tests, including the confirmation of absence of lesions and histopathological effects, has already been conducted by the required regulatory authorities. Repeating such experiments in an academic setting (which is anyway not acceptable by the regulatory authorities) will take several months, and since LDS-AXT is already approved will not be warranted in our opinion.  In addition, all the ingredients composing the nanodroplet formulation are FDA approved and are not known to cause any adverse effects.

Comment: The lager part of nano droplet of LDS-ATX usually accumulates into liver of mice, why the SOD activity of LDS-ATX is less than the control group (Figure4

Reply: It is not known that the large part of the droplets is in the liver. In fact, this is probably not the case. The nanodomains are just vehicles, and as far as we know from decades of research on this type of vehicles (nanodroplets), they reach the membranes of cells lining the digestive truck, do not cross these membranes and, after discharging their bioactive payload (AXT in our case) into the blood, continue in the digestive truck and are excreted in the feces.

Round 2

Reviewer 3 Report (New Reviewer)

Comments and Suggestions for Authors

Thanks a lot for submitting your revised paper to our journal, your work is very amazing and creative, but there are some problems in your paper your need further revised, then we will consider accepting your paper. 

1.  In the title 2.1 and 2.3, it should be VS not V, please change it.

2. The nanoparticle's PDI is so high than 0.5, it means this nanoparticle is unstable, it is hard to play its role in vivo study.

3. How do you remove the unencapsulated ATX during the nanoparticle synthesis process? And does your nanoparticle is stable in the PBS for 7 days? Does this nanoparticle have any size change?

Comments on the Quality of English Language

The English writing is good, just a few problems, please revised your paper again.

Author Response

Thanks a lot for submitting your revised paper to our journal, your work is very amazing and creative, but there are some problems in your paper your need further revised, then we will consider accepting your paper. 

  1.  In the title 2.1 and 2.3, it should be VS not V, please change it.

Corrected

  1. The nanoparticle's PDI is so high than 0.5, it means this nanoparticle is unstable, it is hard to play its role in vivo study.

LDS-AXT is already in the market and shows stability (no change in droplets sizes distribution) of over two years both chemically and physically. The PDI in this specific sample (0.6) is >0.5 but not significantly higher. For instance, please see the following sentence from  the instructions at https://www.malvernpanalytical.com/en/learn/knowledge-center/whitepapers/wp111214dlstermsdefined: “Values greater than 0.7 indicate that the sample has a very broad size distribution and is probably not suitable for the dynamic light scattering (DLS) technique.” A polydispersity of >0.5 therefore might suggest non-thermodynamic stability, but it is not always the case. Many of our systems, such as LDS-CBD, which also has a PDI value of 0.6, have PDI of 0.2 to 0.8 and are still thermodynamically stable. The PDI only means relatively “wider droplets size distribution”. In addition, please see Figure 11, which demonstrates LDS-AXT’s physical stability. We have also previously tested LDS-AXT (which, as said, is already in the market from Jan. 2024) for its droplet distribution at three different temperatures for a minimum of 12 months. Lastly, we don’t understand the claim that droplets with wider droplet size distribution are not playing a role in in vivo studies. What about emulsions consisting of  a very wide droplets distribution? There are several such products currently in the market.

  1. How do you remove the unencapsulated ATX during the nanoparticle synthesis process? 

We are not forming nano solid particles. We rather form nano droplets. The proper term  is “nano entrapment” and not nano encapsulation.  All the droplets are interfacially covered by layers of surfactants and there are no unencapsulated droplets.

And does your nanoparticle is stable in the PBS for 7 days? Does this nanoparticle have any size change?

Yes, the droplets are stable for over two years with practically no droplets sizes changes.

We hope that those clarifications are answering the reviewer questions. The  biovalability and performance results, that are  described in this article, were found also in our other  products already in the market (e.g., CBD based) that have shown similarly fantastic results also on humans for certain indications.  

Round 3

Reviewer 3 Report (New Reviewer)

Comments and Suggestions for Authors

I am agree with that the journal can accept this paper

This manuscript is a resubmission of an earlier submission. The following is a list of the peer review reports and author responses from that submission.

Round 1

Reviewer 1 Report

Comments and Suggestions for Authors

The study investigates the efficacy of a novel nanocarrier in improving the pharmacokinetic profile and antioxidant efficacy of astaxanthin (ATX), a carotenoid nutraceutical known for its poor bioavailability due to high lipophilicity. Key findings of the study include improved bioavailability of ATX, potentially due to its ability to solubilize ATX in an oil-in-water micro-environment, compared to ATX alone, as well as enhanced antioxidant efficacy of LDS-ATX, which was effective in correcting LPS-induced oxidative damage mainly in the liver and brain. Research strengths include using an innovative nano-formulation. The study introduces a novel nanocarrier, LDS-ATX, which demonstrates promising improvements in the bioavailability and antioxidant efficacy of ATX.

Despite the study suggesting that the LDS-ATX nano-formulation significantly improves the bioavailability and antioxidant efficacy of astaxanthin, presenting a promising approach for enhancing its therapeutic potential, the study design and limited analysis mean that this article requires significant additions and is not suitable for publication in its current form.

A few suggestions to improve the manuscript are made here:

1.      In the introduction, it would be beneficial for the authors to include a detailed overview of ATX's antioxidant properties and its potential benefits in mitigating oxidative stress-induced inflammation. That would not only enrich the introduction but also add the significance of the nanocarrier approach investigated in your study.  In addition including information on specific disorders or health conditions where ATX may have shown promise would further contextualize the significance of the study.

2.      For the analysis illustrated in Figures 3 and 4, it would be advisable to utilize two-way ANOVA, allowing authors to compare the results obtained across all investigated tissues.

3.      In the results section: There is no need to mark statistically insignificant differences in the Figures (Figure 3 and 4). Usually, only significant differences are marked.  Additionally, please reconsider changing the group symbols to shorter ones, as the current ones are lengthy and make it difficult to follow the results.

4.      The scope of the in vivo studies appears limited, as only two parameters were determined. It is suggested that additional analyses such as assessing DNA oxidative damage (e.g., 8-hydroxy 2'deoxyguanosine level) and activity of antioxidant enzymes (SOD, CAT, GR, GPx), as well as measuring interleukin levels (IL6, TNF alpha, IL1, IL10) in examined tissues, would enhance the comprehensiveness of the research. Furthermore, given the use of an inflammation induction model, it is recommended to determine the level of C-reactive protein in plasma to confirm the induction of inflammation in animals.

5.      The probable mechanism of action of ATX is associated with the inhibition of NF-kappa B transcription factor activation. This mechanism is associated with the release of pro-inflammatory cytokines, and exploring it would enhance the understanding of ATX's effects.

6.      Two out of the three paragraphs of the discussion lack references to other research, despite providing detailed explanations of oxidative stress definition and measurement methods. It's essential for the authors to discuss their results in reference to existing scientific studies to strengthen the validity and context of their findings.

7.      Overall, the discussion section is lacking in depth. It does not adequately reference other research findings, nor does it present mechanisms to explain the observed results.

8.      The authors fail to address the properties of ATX, such as its fat solubility, which could potentially impact the results obtained in different tissues or parts of the study.

9.        The manuscript omits several crucial methodological details essential for ensuring the study's reproducibility and evaluating its scientific rigor. It lacks information on the diet

10.   of the animals, tissue homogenates preparation, and a comprehensive description of the reagents and equipment, including the names of producers, cities, states (if applicable in the US), and countries. Including these details would significantly enhance the manuscript's clarity and reliability.

11.   The description of the in vivo experiment is incomplete. There is a lack of information regarding the sex, age (or body weight of the mice), procedures before ATX administration, acclimatization period, conditions in the animal facility, and, most importantly, the consent obtained from relevant ethical committees for the research. It's also unclear on what basis the number of animals needed for research was estimated (3 or 5). Additionally, an experimental diagram should be included for clarity.

12.   What was the rationale behind choosing the duration of action of ATX? 24 hours seems relatively short for achieving optimal nutritional levels with the preparation.

13.   The authors should provide complete information regarding the dose of ATX used and justify why only one dose was administered. Additionally, a dose of 10mg/kg appears to be relatively high. Please compare the ATX dose with recomendation

Reviewer 2 Report

Comments and Suggestions for Authors

The article of Kumudesh Mishra et al. “A New Tailored Nano-Droplet Carrier of Astaxanthin Can Improve Its Pharmacokinetic Profile and Antioxidant Efficacy” describes that LDS-ATX is superior to ATX dispersed in oleoresin (AXT) and improves the bioavailability of ATX, reduces oxidative damage to proteins and lipids caused by lipopolysaccharide in the brain, liver and muscle tissues in mice.

The article discusses an important issue, since astaxanthin and its use in clinical practice are underestimated. Undoubtedly, the article should be published, but it needs revision.

There are some comments to authors.

Introduction

1. There is not enough information about astaxanthin and its uses. It is necessary to describe its functions and protective properties, citing relevant studies in the world.

2. You need to describe the use of lipopolysaccharide as an oxidative stress inducer, since you use it in this capacity. Cite relevant studies.

3. Line 58…“levls” replace with ”levels”

4. Figures 1, 2, 3, and 4 are of very poor quality and need to be improved. In the legends to the figures, indicate the number of repetitions of the experiment.

5. The step along the X-axis should be smaller. For example, 0-50-100-150-200

6. Line 65… 2.2. Antioxidant efficacy of LDS-ATX v ATX… What does the letter v mean? Maybe you meant vs?

2.2. Antioxidant efficacy of LDS-ATX vs. ATX

7. Line 68…Why did the authors choose this particular LPS concentration? Justification required.

8. Lines 68 - 71… It is possible to reformulate the proposal in this way… “Subsequently, oxidative stress was induced with 1 mg/kg LPS for 4 hours, and then the extent of pro-oxidant defense in brain, liver, and muscle was determined using commercial protein carbonylation and malondialdehyde kits to determine irreversible protein oxidation and oxidative damage by lipid peroxidation, respectively”... replace with …“Subsequently, oxidative stress was induced with 1 mg/kg LPS for 4h and then the extent of pro-oxidant protection in brain, liver and muscle was determined using protein carbonylation and malondialdehyde commercial kits for respectively determining irreversible protein oxidation and lipid peroxidation oxidative damage.”

9. Line 104…Our unique LDS-ATX formulation, as all other LDS nanodroplet formulations…

What other compositions do the authors not mention? If there are attempts to create nanodroplets in other laboratories, it is necessary to describe this in the text and provide references.

10. Lines 119-122… References required.

11. Lines 122-126… Restate the sentence

12. Lines 122-133… References required.

13. Indicate which astaxanthin the authors used (manufacturer).

14. Figures 7 and 8 are completely unreadable, very poor quality. They should either be significantly improved or excluded from the article.

15. The structure of the article seems unsuccessful to the reviewer. The results section is too short, there is not enough data. Perhaps it should be restructured so that some of the materials from the Methods section are transferred to the results section and appropriate corrections are made in the text.

16. In the discussion section, there are practically no references and mainly only their results are discussed. This section needs to be revised and/or supplemented.

Comments on the Quality of English Language

 Moderate editing of English language required

Reviewer 3 Report

Comments and Suggestions for Authors

1. As Communications, novelty and conciseness should be emphasized. Therefore, is it necessary to list the standard curve in Figure 1 as a general conventional tool?

2. As an original scientific research paper, the data is generally presented through qualitative or quantitative characterization of instrument detection. So what scientific results should Figure 5 express? Is it necessary to display it?

3. Figure 7 and Figure 8 can't be seen clearly.

4. What do the curves of the red and green colors in Figure 7 represent respectively?